# Novel Foods and Sustainability as Means to Counteract Malnutrition in Madagascar

**DOI:** 10.3390/molecules26082142

**Published:** 2021-04-08

**Authors:** Maria Vittoria Conti, Aliki Kalmpourtzidou, Simonetta Lambiase, Rachele De Giuseppe, Hellas Cena

**Affiliations:** 1Laboratory of Dietetics and Clinical Nutrition, Department of Public Health, Experimental and Forensic Medicine, University of Pavia, 27100 Pavia, Italy; rachele.degiuseppe@unipv.it (R.D.G.); hellas.cena@unipv.it (H.C.); 2Department of Public Health, Experimental and Forensic Medicine, University of Pavia, 27100 Pavia, Italy; s.lambiase@unipv.it; 3Clinical Nutrition and Dietetics Service, Unit of Internal Medicine and Endocrinology, ICS Maugeri IRCCS, 27100 Pavia, Italy

**Keywords:** malnutrition, Madagascar, sustainable approach, novel food, *Moringa*, insects

## Abstract

Although the trends of international reports show an increase in overweight and obesity, even in developing countries, there are still areas of the world, such as Sub-Saharan Africa, strongly affected by undernutrition. Specifically, in Madagascar, the percentage of stunted children under 5 is extremely high. Furthermore, the COVID-19 pandemic is expected to increase the risk of all forms of malnutrition, especially in low-income countries, including Madagascar, with serious intergenerational repercussions. This narrative review aims at investigating eating habits and cooking methods of the Malagasy population, addressing sustainable healthy diets through promotion of novel foods. While novel foods are a recent concept, there are data that describe how they may contribute to counteract food insecurity and malnutrition considering context and place. Efforts to promote native, traditional foods as *Moringa oleifera*, an indigenous plant in Asia and Africa including Madagascar, rich in protein and micronutrients, as well as edible insects, alternative sustainable source of protein, lipids, iron, and zinc, would provide not only nutritional but also cultural and economic benefits. The potential synergies between food traditions and agroecology have the potential to impact health addressing larger issues of sustainability and food security. Regional, national, and international policies are needed to develop and support one health approach actions.

## 1. Introduction

The term ‘malnutrition’ has been linked to hunger and undernutrition so far, but it is no longer the case since it may take different forms with diverse health consequences depending on whether it manifests as under- or over-nutrition. 

According to the 2020 WHO definition [1] malnutrition is typically classified into three different forms: (i) undernutrition (wasting, stunting, underweight); (ii) micronutrient deficiencies, i.e., “hidden hunger” (micronutrient deficiencies—MND); (iii) over-nutrition such as obesity and overweight (major risk factor for non-communicable diseases—NCDs) [2].

Classification of malnutrition is an attempt to better capture the consequences on nutritional status of dietary patterns modification over recent decades [3]. Identification of MND as a separate category, was intended to classify those conditions arising from poor quality diets and related micronutrient inadequacies despite energy adequate intake [3]. The ‘over-nutrition and obesity’ category was then provided to broaden the concept of malnutrition ranging from undernourishment to excessive nutrient and/or energy intake and to capture the rapid increase of overweight/obesity and NCDs [4]. In 1990s, the concept of the double-burden of malnutrition, now triple-burden of malnutrition, was introduced to highlight the co-existence of these forms of malnutrition within countries, communities, families and even within individuals [3]. Describing the growth trend related to the current global situation, UNICEF has reported [5] how obesity prevalence and related pathologies are increasing not only in high-income countries, but also in middle- and low-income ones. In fact, in continents such as Africa or Asia the triple burden of malnutrition is increasingly apparent [5]. The raising prevalence of these three conditions of malnutrition and their coexistence, even in developing countries is strongly associated with socio-economic aspects. Due to globalization, the availability of processed and low-cost food, affordable and easily accessible by local population, has increased in African food systems [6].

Despite the increasing alert of international reports and attention to the U.N. Sustainable Development Goals (SDGs) malnutrition rates remain alarming [5,7,8,9]. Stunting is declining too slowly, while wasting still impacts too many children [7]. According to the Food security and nutrition report (FAO), Africa is the only region in the world where the number of stunted children has risen, shifting from 50 to 58 million of stunted children under the age of 5 [5,8].

Although the trends of international reports show an increase of all three forms of malnutrition and their coexistence in developing countries, there are still areas strongly affected by undernutrition. In Madagascar, for example, the percentage of stunted children under 5 is extremely high (>30%), and the percentage of overweight children is still classified as “low” (2.5–5%) [5].

According to the 2020 Global Hunger Index (GHI), Madagascar is one of the three countries with an alarming level of hunger, along with Chad and Timor-Leste [10].

Additionally, the COVID-19 pandemic has undermined food and nutrition security for many, and these effects will likely increase into the future [10]. The measures taken throughout the world to contain COVID-19 spread have already increased food insecurity by limiting access to fields and markets in some areas, raising food prices with local spikes and reducing income-earning opportunities, thereby limiting the ability of vulnerable populations to purchase food [11]. Moreover, the pandemic is affecting nutrition and specifically, schools have been closed at various points in 2020, preventing access to nutritious meals for children in many cases [5,12].

Early in COVID-19 pandemic, UNICEF estimated a 30% overall reduction in essential nutrition services coverage, reaching 75–100% in lockdown contexts, including fragile countries with humanitarian crises [13].

As anticipated, given the current trajectory, the goal of achieving zero hunger by 2030 will not be fully achieved, as was already envisaged before COVID-19 pandemic [12]). According to the World Food Programme [9] and a *Lancet* report [13], the number of people in low- and middle-income countries (LMICs) facing acute food insecurity will nearly double, reaching 265 million by the end of 2020 [5,12].

COVID-19 pandemic is expected to increase the risk of all forms of malnutrition with serious intergenerational repercussions. Specific actions on early life nutrition should be integrated into country strategies to avoid unfortunate fallout on child growth and development as well as chronic disease containment, and overall human capital formation [13].

Focusing on Madagascar, Malagasy Ministry of Agriculture and Ministry’s Regional Direction for Analamanga (City of Antananarivo), aware of the current condition, have built a consortium to operationalize the City Region Food Systems (CRFS) approach proposed by FAO and RUAF partners (the International Water Management Institute (IWMI)) and the Laurier Centre for Sustainable Food Systems). This approach was committed to create regional food strategies, and to design and implement a “post-COVID-19” strategy, which may serve as an effective tool to reinforce food system and increase resilience [14].

Considering the emerging situation in low-income countries, as in the case of Madagascar, this narrative review aims at investigating eating habits and cooking methods of the Malagasy population addressing sustainable healthy diets through promotion of novel foods, with a focus on insects, to prevent malnutrition. The topics treated are summarized and presented graphically in Figure 1.

## 2. Food Habits in Madagascar

As described above, Madagascar is one of the three countries with an alarming level of hunger and is one of the 20 countries, all over the world, with the higher prevalence of undernutrition [15].

The national prevalence of under-five stunting is 50%, which is significantly higher than the 25% mean rate in developing countries. In Madagascar, 50% of infants under 6 months are exclusively breastfed, which is well below the Eastern Africa average of 60% [16]. The Madagascar adult population also faces a malnutrition burden: 37% of women of reproductive age are affected by iron deficiency anaemia, while 6% and 5%, respectively, of adult men and women are affected by diabetes mellitus type II. Meanwhile, 7.5% of women and 3% of men are affected by obesity [16].

In low-income countries population’s dietary habits, strictly connected with health status, mainly depend on food security, which refers to stable food availability and access. Food insecurity affects a high proportion of the population in Madagascar, especially in rural areas [15]. In this context of food insecurity, there is a lack of information related to dietary guidelines that must be transposed as literally as possible to ensure respect of the nutritional requirements and counteract malnutrition [17].

Ravaoarisoa et al. conducted a cross-sectional study in the Amoron’i Mania region to investigate food habits of non-pregnant mothers with children [15]. The authors used a 24-h recall method for mothers’ dietary diversity assessment and a food frequency questionnaire for measuring dietary intake during the post-harvest period. According to the 24-h recall, rice was the most consumed food item (by 100% of the sample), followed by sweet potato, cassava, and potatoes. Indeed, rice is the staple food for most Malagasy and is considered a “king product” [18].

Similar results were described also by Randrianarison et al. [18] reporting that food consumption, in Atsimo Atsinanana Region in south-eastern Madagascar, was characterized by a high priority for staple foods, so that three quarters of the daily energy intake was covered by rice and cassava. Green-leafy vegetables of “anandrebaka”, “petsay” and sweet potato leaves were part of their daily diet, on the contrary, legumes, fish, zebu meat and pork meat were less frequently consumed, as well as fruit and dairy products [15]. Indeed, meat consumption was low and the proportion of mothers who had never consumed dairy products, milk and eggs was high [15]. Despite the high biodiversity of the region people did not follow a balanced diet [18].

Ravaoarisoa et al. confirmed that in the Amoron’i Mania region, pregnant and breastfeeding women had low diet variety, with a rather monotonous pattern, poor in fruit, vegetables, and protein [19]. Moreover, dietary habits were influenced by many factors, including agricultural products grown in the region, their availability during the year (self-consumption), purchasing power (in case of shortage) and tradition [19].

Regarding fortified products, iodized salt was used every day by a small percentage of the sample (10% of the mothers). It was used irregularly by 19%, never by 31% while 40% of mothers did not know if the salt they used was iodized [15].

Low dietary diversity score (<5) was significantly more common among farming mothers, with low education level as already reported by other authors in sub-Saharan regions [15].

Despite the high frequency of energy dense food consumption, data available at national level showed a poor energy intake (less than 2250 kcal per day) in 58% of Malagasy adults in 2013. In 60% of the population, carbohydrates (rice, sweet potato, cassava, etc.) provided 85% of calories, instead of 55–75% as recommended by WHO [20]. Furthermore, vegetables consumption remains low except for green leaves varieties with longer growing seasons [15]. In general, subjects consumed the vegetables varieties grown in the fields during the study period and 58% of the mothers consumed the “anandrebaka” seed, easily found in the fields [15].

Reuter et al. also investigated wild meat consumption reporting a high intake by the indigenous population. Bats and tenrecs (a diverse collection of mammal species endemic to Madagascar) were the most consumed animal sources, and people with a stated preference for bats and wild pigs consumed them with the same frequency (two times per week) as for chicken and domestic pig [21].

## 3. Traditional Cooking

In Madagascar, food has always been cooked using simple methods and techniques, such as roasting over a fire, grilling over hot stones or coal and boiling, first in containers made of green bamboo, then in clay pots and metal vessels. Food preservation techniques include smoking, sun-drying, and salting [22].

The use of these specific cooking methods is not entirely positive for health in fact, biomass burning is a major contributor to household air pollution, which may exert negative impact on respiratory and cardiovascular health of primarily women and children [23].

Smoked dried beef, kitoza, salted dried fish and many other foods, are still prepared in a similar way [22]. The process of fermentation was adopted early in the history of Malagasy cuisine to produce curds from milk, to amplify taste of some fresh and dried tubers and to make alcoholic drinks [22].

Madagascar cuisine is charming in its simplicity. A traditional Malagasy dish features an unusual large amount of rice with a modest portion of chicken or fish, sauce, and vegetables. There is a popular belief that a meal without rice will lead to a sleepless night. The presence of rice in contemporary cuisine across the entire island is a constant; however, farmers in the arid south and west often substitute rice with corn, cassava or curds made from fermented zebu milk. A cup of rice alone is considered a very acceptable meal. Red rice for breakfast is ubiquitous throughout the country, often cooked with extra water, producing a soupy rice porridge, known as sosoa [22].

As for meat, the principal source remains zebu cattle. The Malagasy people enjoy better cuts served as zebu steaks or zebu stew, while less tender pieces, cut into small cubes, are boiled until very tender in salted water along with garlic and onions, shredded and finally roasted [15].

Islanders also consume chicken and goat; but pork, although available, is taboo in many areas of the country because of religion [15]. Vegetables are served simply boiled or with spices to boost flavour. The other primary element of a traditional Malagasy dish is sauce. Sauces vary by region and are flavoured with onion, garlic, tomato, ginger, curry, vanilla, coconut and, at times, herbs, and spices [15].

Generally, tomato-based sauces are used in the highlands and coconut-based sauces in the coastal regions [15].

Flavourings vary significantly from dish to dish and from region to region. Coconut-based seafood and crustaceans are frequently consumed in the northeast coastal region; thyme, basil and lemongrass-scented dishes in the central plateaus; clove, pepper, cinnamon, ginger and lemongrass-infused recipes in the eastern part of the island; kaffir lime and lemon-flavoured specialties in the drier western areas and in the southern regions; finally, dishes infused with aromatic plants in the northern regions along the Antsiranh and Amber mountain ranges [22].

The term “bredes” refers to plants and vegetables that are classified in two categories defined by flavour. Brides are usually mild leaves while anamolaho are spicy.

Nevertheless, the food in Madagascar exhibits delicious tastiness without being hot and spicy. Those preferring extra zip in their rice or toppings cautiously add touches of chili relish (satay) that comes in strengths from hot to fiery spicy, or spicy mango, vinegar-preserved carrots, lemon pickles or hot curry oil. These toppings are served separately from the food and are added accordingly to personal taste [22].

## 4. Sustainable Solutions and Novel Food

Regarding the public health nutrition framework in Madagascar, characterized by a difficult socio-health condition, practical examples of sustainable approach, by utilization of territorial resources are essential. According to the Barilla Center for Food and Innovation and *The Economist* report [24], “good practices” will require new approaches for sustainable agriculture, focused not only on a greater yield of agricultural activities, but also on better access to food resources and on reduction of environmental impact of agriculture, of greenhouse gases production [24].

What is good for human health is often also good for the planet. This means that consumers’ food choices can have an impact on the environmental sustainability of our food supply. The World Resources Institute (WRI) has indicated some changes in eating habits that could positively affect human health, while reducing the need for agricultural land and reducing greenhouse gas emissions choosing foods like vegetables, instead of animal products [24].

Efforts to promote native, traditional foods as Moringa oleifera, may provide not only nutritional but also cultural and economic benefits, highlighting that potential synergies between food traditions and agroecology have the potential to impact health addressing larger issues of sustainability and food security [24].

## 5. Novel Foods

Novel food is defined as “*food that had not been consumed to a significant degree by humans in the EU before 15 May 1997*”, when the first regulation on novel food came into force [25]. “*Novel food can be newly developed, innovative food, food produced using new technologies and production processes, as well as food which is or has been traditionally eaten outside of the EU*” [25]. Examples of novel food include new sources of vitamin K, extracts from existing foods, agricultural products from third countries such as chia seeds, noni fruit juice, or food derived from new production processes [25]. Although it is still a recent concept, there are data that describe how novel foods may help to alleviate the problems of poverty and food insecurity, but only if steered by continual policy development and actions at the regional, national, and international level [25]. For instance, the great progress made with iodized salt for preventing iodine deficiency disorders, through global partnership, provides inspiration for future applications of nutritional science and food technology to public health problems in the developing world [26]. The economic development that has led to improved food security and better health in some countries needs to be harnessed, while at the same time actions to avert the adverse health effects of the nutrition transition need to be taken. Consequently, the potential of novel foods to alleviate undernutrition is becoming more apparent [26].

### 5.1. Moringa

Moringa is a plant native to Asia and Africa, including Madagascar [27]. Moringa oleifera (*M. oleifera*) is the most known and used species of the family Moringaceae due to its beneficial health effects and its multiple uses in food, cosmetics, medicine, forage for livestock and water purification [28]. The properties of the other family species remain largely unknown and unstudied [28]. Moreover, *M. oleifera* could be easily cultivated in any tropical and subtropical region [29]. All parts of *M. oleifera* can be used and each part has different pharmacological actions [28], however leaves are the most used [27]. *M. oleifera* leaf powder is significantly richer in vitamin C, vitamin A, calcium, iron, potassium, and protein than other commonly consumed foods like oranges, carrots, milk, spinach, banana, and yogurt, that are known for high concentration of these specific nutrients [29,30]. Multiple studies have been conducted about the beneficial effect of *M. oleifera* leaf powder on haemoglobin concentration to counteract anaemia mainly in children [30,31,32,33,34] and women of reproductive age [35,36,37,38] in Africa and Asia. Furthermore, *M. oleifera* has been used traditionally for the management of diabetes and the maintenance of glycaemic control, however the scientific evidence until now derives mainly from in vitro and animal studies [39,40].

The presence of phytate in raw *M. oleifera* leaves and seeds reduces the bioavailability of specific nutrients, so the processing of the *M. oleifera* leaves and seeds is necessary in order to maximize intake of these nutrients from *M. oleifera* [29,41,42]. As a result, processed *M. oleifera* seed flour is an alternative source of nutrients [29,42]. Moreover, *M. oleifera* leaves can be preserved for a long period of time without losses of nutrients and consequently it could be considered an ideal tool against malnutrition in many low- and middle-income countries, such as Madagascar [29].

Although *M. oleifera* is an indigenous plant in countries like Madagascar, there is limited acknowledgment by the local population of its potential use and beneficial effects. *M. oleifera* is potentially a sustainable and affordable solution against malnutrition in Madagascar, however its consumption is low, and locals likely consume other leafy vegetables for stocks rice recipes [43,44,45,46].

According to Rakotosamimanana et al. Malagasy locals habitually eat mostly rice or cassava depending on the season, adding sugar or coconut [46]. Addition of leafy vegetables in the meals is not so common. In the southern part of Madagascar locals more often consume *M. oleifera* and cultivate it in their gardens, showing a greater awareness about its health properties compared to the people who live in the central part on the country [46]. Price and availability of *M. oleifera* in local markets play a key role, since cassava was consumed frequently in all the regions. Additionally, in the south part of Madagascar, *M. oleifera* leaves are consumed more frequently salted while in the central part with oil. However, *M. oleifera* leaf powder added to snacks based on cassava did not decrease liking [46]. Even though *M. oleifera* is very nutritious its habitual use is limited due to the unpleasant bitter taste of *M. oleifera* leaf powder, which becomes even stronger after drying procedure [47]. For this reason, it could be used as a food ingredient for home fortification or as fortifier ingredient by food companies for products consumed by local populations, including cereals, rice, floor, pasta, tablets/capsules, juices, snacks, sauces, and commercial broth cubes used in staple meals [47].

Zahana, which is a local organization in Madagascar, has created actions for cultivation and use of *M. oleifera* locally, being a low-cost option for nutritional improvement in their villages. For this reason, school-teachers were invited to plant and eat *M. oleifera* at school, given their role as reference for the children helping them to get acquainted with *M. oleifera* trees and familiar to its taste. In children it is easy to notice an increase liking of bitter taste foods by mere exposure, especially pairing novel or disliked foods, like bitter vegetables, with liked flavours repeatedly [48]. Moreover, both school and family have a particularly high-standing role in what is called ‘taste education’ process [48]. In this context, in 2019 the Zahana organization offered free meals in schools accompanied by *M. oleifera* leaf tea for children. Additionally, they planned to cultivate a forest of *M. oleifera* using the best variety of *M. oleifera* seeds available in Madagascar as a sustainable approach to counteracting rural malnutrition [49,50].

### 5.2. Edible Insects

Even though entomophagy (from Greek ἔντομον -éntomon, that is “insect” and φᾰγεῖν -phagein, that is “to eat”) is not common in the western world, insects are consumed by around two billion people globally [51]. They have been a source of food for human population over the years and they are consumed highly in Africa Asia and Latin America [52]. The exact number of insect species used as food is still unknown [53], but it is estimated around 1900 [54] and 2300 according to different authors [55,56]. Lately, industries and academies in Europe and the USA have started to show interest in insects as food source [57]. The “disgust” factor, availability and price remain important reasons for the acceptance of edible insects, however the popularity and the sales of insect-based food products are increasing also in these countries [57]. Many people refuse insects as food for cultural reasons (psychological, social, religious, anthropological reasons, etc.), therefore a change in eating habits is not expected to occur fast. Moreover, most Western people refuse to eat insects because they consider them dirty and contaminated. However, most edible insects, such as grasshoppers, locusts, butterfly larvae and beetles, are herbivores and consequently more hygienic than crabs or lobsters, which eat carrion and are sometimes harvested from polluted aquatic systems [57,58]. This attitude is therefore a form of prejudice, overcome in case of other organisms such as frogs and lobsters once considered inedible. History reports that, giving to eat lobster to the servants was considered a cruel punishment whereas nowadays lobster is considered a delicacy only for refined palates. So, promoting entomophagy is emerging and perhaps inevitable, but the question is: how fast will the acceptance process be? [58]. A few of years ago, for example, in Belgium, ten species of insects were allowed for sale: *Acheta domesticus, Locusta migratoria, Zophobas atratus morio, Tenebrio molitor, Alphitobius diaperinus, Galleria melonella, Schistocerca americana gregaria, Gryllodes sigillatus, Achroia grisella* and *Bombyx mori* [59]. Most edible insects are terrestrial [58], and they belong to the following orders *Ortoptera, Coleoptera, Lepidoptera, Emiptera, Isoptera*; they can be eaten at the larval or adult stage according to the taxon. The importance of edible insects for feeding and security has already been highlighted [54,60]. In fact, edible insects are considered a new alternative sustainable source of proteins that exhibits higher feed-conversion efficiency and less negative environmental impact, compared to conventional animal-derived protein sources [60]. Human population is growing fast with an estimation of 9 billion earth habitants in 2050 and a growing need of food. Highly nutritional edible insects may help solve issues of present and future global protein requirements, food insecurity and malnutrition [57].

Edible insects offer high-quality animal-derived protein [57], though depending on the species and the stage of life [54,61,62], and are also good sources of lipids, fiber, vitamins and minerals, including iron, and zinc [54].Protein quality for humans is measured by amino acid profiles and digestibility. In Nigeria four popular edible insect species have been shown to contain all essential amino acids, with relatively high amounts of lysine, threonine, and methionine, which are the major limiting amino acids in cereal- and legume-based diets [62]. The saturated/unsaturated fatty acid ratio of most edible insects is 40% lower than poultry and fish, but the content of polyunsaturated, linoleic, and linolenic acids, is higher [63]. Although amino acid profiles differ among edible insect species [64] most of them have amino acid levels that are comparable to beef, egg, and soy or even higher in some cases. In some African countries (Angola, Kenya, Nigeria and Zimbabwe), where corn is the most common staple food, there is a shortage of tryptophan and lysine, and the diet is well supplemented by termite species such as *Macrotermes bellicosus* or *Macrotermes subhyalinus* rich in AA [65]. Noteworthy is also the fat composition of edible insects, which varies from 10–60% of dry weight among species and their developmental stage, with the highest values reported in termites and palm weevil larvae [66] as well as in immature stages.

Regarding fiber, chitin is present in high concentrations in insects, ranging from 11.6 to 137 mg/kg of dry weight [67]. Even though chitosan is indigestible, it has a key role in parasitic infections and allergies [68,69]. Although the greatest attention is given to proteins and unsaturated fats, insect micronutrient composition may help fulfil requirements and positively impact immune system as well as physical and mental growth impairment in developing countries [66].

In Africa, mainly in central Africa, around 470 kinds of insects are consumed and, entomophagy is common among local populations [54]. In Kinshasa, the average consumption of caterpillars per household is around 300 g [70], while in the Democratic Republic of Congo around 15% of the protein intake derives from insects and in the Central Africa Republic, people who live in the forest may find only insects as their main source of proteins [60].

In Madagascar, entomophagy is part of the culinary traditions. According to Randrianandrasana et al. (2015) [70], 53 edible insects have been reported be commonly consumed in rural areas of Madagascar, depending on the availability and the seasonality of edible insects per area [70]. The most consumed insects were *Coleoptera* and specifically adult beetles, second in the rating were Hemiptera (true bugs) mainly terrestrial ones consumed both in immature and adult stages, followed by *Lepidoptera* (butterflies and moths) consumed mainly in the pupal stage and *Orthoptera* (grasshoppers, locusts and crickets) [70,71,72]. A small number of participants reported the consumption of *Odonata*, mostly large dragonflies, and *Hymenoptera*, mostly honeybees and vespid wasps [70]. Edible insects are usually collected by farmers from the fields or forests in the rural areas, but they are also sold in local markets of urban areas. The prices of edible insects are low during the high season and they increase when the supply is low [71]. The general increase of the prices during the last years could be explained by the lower availability of edible insects in those areas [71]. When they are collected in small numbers, they are usually grilled over fire. The internal organs and/or legs are removed from the larger insects and the heads are removed from beetles. When insects like locusts are gathered in larger numbers, they are usually boiled or sundried [71]. Sometimes insects are cooked with extra herbs and vegetables like onions and tomatoes and other insects like locusts are served as side dishes of meat main dishes [70].

According to Dürr et al. [71], the locals in the central highlands of Madagascar prefer edible insects instead of meat due to their more affordable price and their taste [71]. The highest availability of edible insects was reported from October to December which is a period between exhaustion of rice supply and rice harvest [70]. Consequently, farming and storage of edible insects, in different ways, could be an important sustainable source of protein during famine or poor harvest-end periods offering a secure and cheap supply of nutritious food. Besides the nutritional value, noteworthy insect farms can arise anywhere [63], but this brings up another issue, food safety and hygiene techniques promotion. Information to the local populations about edible insects, is needed, and compliance with regulatory obligations in their value chain, is required [73].

Studies have shown a potential food allergy risk related to edible insects’ consumption [74]. Pesticides residues, mycotoxins, antinutrients, pathogens, heavy metals and parasites are health risk factors that need to be considered [73].Edible insects can be used as food ingredients in the form of powder in widely accepted food products, like bread, snacks, pasta, bars, muffins, juices, etc. Currently, cricket powder has become well-known and accessible in most parts of the world and the acceptability of products containing cricket powder has increased even in western countries [75,76,77,78]. For example, bread and muffins enriched with 10% of cricket powder have shown good technological properties and higher nutritional value than the common bread products without reducing their sensory attractiveness [76,78]. Additionally, according to Kim et al. [79] cricket powder can reach a shelf life of 18 months at room temperature without changes in quality and content characteristics. According to Alves et al. [80] beetles could be good candidates to produce oil due to their high concentration of unsaturated fatty acids and low toxicity as shown in oils extracted by specific species of beetles (*Tenebrio molitor* and *Pachymerus nucleorum*) [80]. Food products containing fermented edible insects like pastes, sauces, etc. could be a solution for the promotion of food security in Africa according to Kewuyemi et al. [81], due to the induction of antimicrobial, nutritional and therapeutic properties by conducting the specific fermentation process [81]. Edible insect powders and oils could be a good solution for fortification of commonly used food products like rice in Madagascar.

In conclusion, the use of alternative food sources with lower water and carbon footprint has become demanding [82]. The development of innovative technologies and food products with the aim of increasing consumption of edible insects may have a key role proposing diets with greater nutritional value to combat malnutrition in developing countries in a more sustainable way.

## 6. Conclusions

In summary, although recent data show an increase of overweight and obesity worldwide even in developing countries, there are still areas of the world strongly affected by undernutrition. Madagascar is one of the developing countries in which the undernutrition condition still represents a huge health concern. These forms of malnutrition represent a major impediment to achieve sustainable development, with crippling consequences for human health, environment, and human capabilities. While there are multiple underlying determinants of malnutrition, sub-optimal diets serve as a common factor for poor nutrition outcomes. Knowing that dietary habits, healthy or not, are an outcome of food systems, it is important to understand how food systems are changing and their ability to deliver nutritious diets while at the same time minimizing negative environmental impact. The identification of dietary patterns that are beneficial for both humans and the environment health is the solution. As described above Madagascar is a place rich in natural cheap and available food sources, for the local population. *M. oleifera* leaves and insect consumption instead of farmed or farmyard animal products or intensive cultivation of cereals (such as rice which is often imported from India) could improve dietary quality and diversity of the diet, conferring great environmental benefits, with significant reductions in greenhouse gas emissions, water, and land use. Therefore, the government’s role is fundamental and food policy interventions are essential, aiming to: (i) improve dietary diversity; (ii) inform the population about the role of sustainable diets for health and environment by using local products; (iii) train the population about cooking methods useful to achieve the highest nutrient bioavailability and ensure food safety; (iv) act through fortification programs that involve the use of moringa and insects powder, within local recipes and habits; (v) promote local products consumption with high nutritional value.

## Figures and Tables

**Figure 1 molecules-26-02142-f001:**
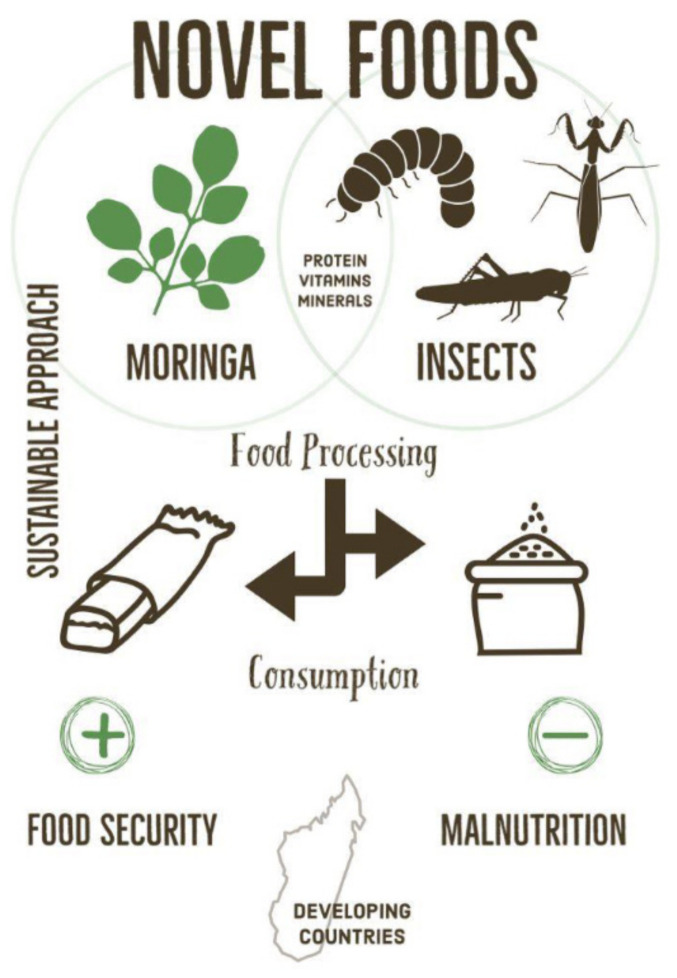
The figure summarizes what the manuscript wants to communicate: how novel foods—specifically *Moringa* and insects in this case—could represent a sustainable approach to counteract malnutrition in developing countries.

## Data Availability

Not applicable.

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
