# Peer review of "Novel Foods and Sustainability as Means to Counteract Malnutrition in Madagascar"

_molecules, 2021, doi:10.3390/molecules26082142_

Round 1

Reviewer 1 Report

Some minor comments are listed below:

Scientific names of plants or insects should come in italics. Please, revise it along the manuscript.

“In vitro” should be also written in italics. Please, revise it along the manuscript.

Please, revise punctuation, double spaces, lack of spaces along the manuscript.

Names of minerals should come in lower case letters. Please, revise it along the manuscript.

Please, explain what “AA” refer to, when firt menctioned (amino acid).

Line 252-255. Please, include the amount (g/100) (fresh or dry weigh, specify it) of the specific nutrients highlighted for Moringa in these lines.

Author Response

The Authors are very grateful to the Reviewers for the useful comments and suggestions.

Dear Reviewer 1,

following you can find the responses to your comments:

  1. Scientific names of plants or insects should come in italics. Please, revise it along the manuscript. The authors thank the reviewer for the comment and corrected the mistackes along the text.
  2. “In vitro” should be also written in italics. Please, revise it along the manuscript.The authors thank the reviewer for the comment and corrected the mistake along the text (line #261).
  3. Please, revise punctuation, double spaces, lack of spaces along the manuscript.The authors thank the reviewer for the comment and corrected all the mistakes related to punctuatuion, double spaces, lack of spaces along the text.
  4. Names of minerals should come in lower case letters. Please, revise it along the manuscript. The authors thank the reviewer for the comment and corrected the mistake. The minerals are now written in case letters (lines #25, #256 and #341).
  5. Please, explain what “AA” refer to, when firt menctioned (amino acid).The authors thank the reviewer for the comment and replace the "AA" with amino acid (line #350).
  6. Line 252-255. Please, include the amount (g/100) (fresh or dry weigh, specify it) of the specific nutrients highlighted for Moringa in these lines. The authors thank the reviewer for the comment and they specify that the quantity is referred to the leaf powder - dry weigh (line #256).

Please see the attachment the final version of the manuscript revised. 

Reviewer 2 Report

The authors reviewed the current situation of malnutrition in Madagascar. They also presented Malagasy cuisine, consumption habits and food preparation with shortcomings and deficiencies. The authors also are presenting the novel foods option, while the Sustainability topic is discussed. Overall I liked the article and I consider that it deserve publication, because the topic is a sad story that must be said.

However, I would have liked to see it more illustrated (including if possible more figures), possibly summarizing some results in a table. Eventually a table showing the insects consumed, the preparation method, the nutritional component, the bibliographic references, etc. could be introduced.

As a result of my evaluation I have additionally some minor corrections. Please see my suggestions below.

After line 101 - The number of the figure is missed, and the figure was not referenced in the text.

The references are not in the journal style neither in the text, not in the reference list.

Lines 107-108 – please mention the other 20 countries with the higher prevalence of undernutrition, and the other 2 countries with an alarming level 107 of hunger

The plant species should appear in italics, i.e. Moringa oleifera (M.oleifera). Also keep a space between M. and olifera (i.e. M. olifera). Please check and correct in the whole manuscript

Line 264 – please keep a space in front of the references.

Line 338 – keep a space in front of “Protein”

Line 343 – keep a space in front of the reference

Line 355 – keep a space after dot

Line 365 – a space is missed also between the brackets

Line 383 – a space is missed also between the brackets

Line 397 – keep a space after the dot

Line 400 – keep a space in front of 10%

Line 409 – space between the brackets

Author Response

The Authors are very grateful to the Reviewers for the useful comments and suggestions.

Dear Reviewer 2,

following you can find the responses to your comments:

  1. After line 101 - The number of the figure is missed, and the figure was not referenced in the text. The authors thank the reviewer for the comment and added the figure number ("Figure 1"); moreover the authors mentioned the Figure 1 in the text (lines #101-#102).
  2. The references are not in the journal style neither in the text, not in the reference list. The authors thank the reviewer for the comment and corrected the references both in the text and in the reference list.
  3. Lines 107-108 – please mention the other 20 countries with the higher prevalence of undernutrition, and the other 2 countries with an alarming level 107 of hunger. The authors thank the reviewer for the comment and added a list of the 20 countries all over the world with higher undernutrition prevalence (lines #109-111).
  4. The plant species should appear in italics, i.e. Moringa oleifera (M.oleifera). Also keep a space between M. and olifera (i.e. M. olifera). Please check and correct in the whole manuscript. The authors thank the reviewer for the comment and corrected the mistackes along the text.
  5. Line 264 – please keep a space in front of the references. The authors thank the reviewer for the comment and corrected it adding a space in front of the references (line #265).
  6. Line 338 – keep a space in front of “Protein”. The authors thank the reviewer for the comment and corrected it adding a space in front of "Protein" (line #337).
  7. Line 343 – keep a space in front of the reference. The authors thank the reviewer for the comment and corrected it adding a space in front of the reference (line #345).
  8. Line 355 – keep a space after dot. The authors thank the reviewer for the comment and corrected it adding a space after the dot (line #355).
  9. Line 365 – a space is missed also between the bracket. The authors thank the reviewer for the comment and corrected it adding a space between the brackets (line #365).
  10. Line 383 – a space is missed also between the bracket. The authors thank the reviewer for the comment and corrected it adding a space between the brackets (line #383).
  11. Line 397 – keep a space after the dot. The authors thank the reviewer for the comment and corrected it adding a space after the dot (line #397).
  12. Line 400 – keep a space in front of 10. The authors thank the reviewer for the comment and corrected it adding a space in front of the 10 (line #403).
  13. Line 409 – space between the brackets. The authors thank the reviewer for the comment and corrected it adding a space between the brackets (line #409).

Please see the attachment the final version of the manuscript revised. 
